# Cross-Linking Reaction of Bio-Based Epoxy Systems: An Investigation into Cure Kinetics

**DOI:** 10.3390/polym16172499

**Published:** 2024-09-02

**Authors:** Pietro Di Matteo, Andrea Iadarola, Raffaele Ciardiello, Davide Salvatore Paolino, Francesco Gazza, Vito Guido Lambertini, Valentina Brunella

**Affiliations:** 1Department of Chemistry, Università di Torino, Via Pietro Giuria 7, 10125 Turin, Italy; pietro.dimatteo@unito.it; 2Department of Mechanical and Aerospace Engineering, Politecnico di Torino, C.so Duca degli Abruzzi 24, 10039 Turin, Italy; andrea.iadarola@polito.it (A.I.); raffaele.ciardiello@polito.it (R.C.); davide.paolino@polito.it (D.S.P.); 3Materials Technical Expertise, Centro Ricerche Fiat (Stellantis), C.so Giovanni Agnelli 220, 10135 Turin, Italy; francesco.gazza@crf.it (F.G.); vitoguido.lambertini@crf.it (V.G.L.)

**Keywords:** bio-based, curing, differential scanning calorimetry, epoxy resin, kinetic analysis, modelling

## Abstract

The cure kinetics of various epoxy resin mixtures, comprising a bisphenol epoxy, two epoxy modifiers, and two hardening agents derived from cardanol technology, were investigated through differential scanning calorimetry (DSC). The development of these mixtures aimed to achieve epoxy materials with a substantial bio-content up to 50% for potential automotive applications, aligning with the 2019 European Regulation on climate neutrality and CO_2_ emission. The Friedman isoconversional method was employed to determine key kinetic parameters, such as activation energy and pre-exponential factor, providing insights into the cross-linking process and the Kamal–Sourour model was used to describe and predict the kinetics of the chemical reactions. This empirical approach was implemented to forecast the curing process for the specific oven curing cycle utilised. Additionally, tensile tests revealed promising results showcasing materials’ viability against conventional counterparts. Overall, this investigation offers a comprehensive understanding of the cure kinetics, mechanical behaviour, and thermal properties of the novel epoxy–novolac blends, contributing to the development of high-performance materials for sustainable automotive applications.

## 1. Introduction

Epoxy resins have become key materials in a wide range of industrial sectors, including automotive and aerospace, thanks to their unique combination of mechanical, thermal, and chemical properties [1]. Nevertheless, the prevalent use of Di-Glycidyl Ether of Bisphenol A (DGEBA) in their production, an agent recognised as a reprotoxic [2], raises critical concerns.

In light of this issue, scientific research has increasingly focused on the development of bio-based epoxy systems as a sustainable alternative to petroleum-derived counterparts. In particular, the use of bio-based resins is taking on a particularly interesting position within the field of composites [3].

This surge of interest underlines the nature of this study, which seeks to build novel epoxy materials through the amalgamation of different bio-based epoxy systems at varying concentrations with relevance to the automotive sector, where we envisage their use. It would offer an appealing avenue oriented towards the replacement of conventional epoxy systems with less toxic and eco-friendly alternatives.

Also, European legislation [4] to reduce carbon emissions is driving the automotive industry towards the use of green materials.

The development of novel materials using this strategy represents an approach that not only facilitates the increase in the amount of biologically sourced constituents but also allows the achievement of good thermal and mechanical properties in the final product [5].

Among the various bio-based alternatives, Cashew Nut–Shell Liquid (CNSL) is regarded one of the most promising materials due to its characteristics including excellent water and chemical resistance, a wide range of pot life and glass transition temperature, good reactivity at room temperature, high biological content, good processability, and low cost [6].

Based on our previous studies, which demonstrated competitive tensile properties in cardanol-based compounds [7], this work focuses on different formulations to enhance the balance between bio-content and mechanical performance meeting specific demands of the automotive industry.

Diverse commercial products have been blended: a bisphenol-type epoxy resin with a CNSL-based diluent, a multifunctional epoxy–novolac resin, a CNSL-based difunctional glycidyl ether epoxy resin, and two amine hardeners reaching a final bio-content of up to 50%.

However, the production of new epoxy systems, combining various commercially available ones, requires a comprehensive study of the curing process: this is essential because the interaction between distinct components can give rise to unknown kinetics. Therefore, when the material composition is unknown or the reaction process is exceptionally complex, the adoption of phenomenological models becomes necessary, as exemplified in this study.

Consequently, the curing kinetic analysis assumes relevance in understanding the final mechanical, thermal, and chemical properties. Specifically, it allows the determination of the characteristics of the cross-linking process, which can be affected by many factors, such as the composition of the reactive mixtures, the curing temperature and time [8].

Furthermore, kinetic analysis enables the determination of a set of kinetic parameters. These data are an essential tool in the prediction process optimising system behaviour under a wide range of temperature profiles.

There are two different approaches to obtain detailed information on curing behaviour by Differential Scanning Calorimetry (DSC) [9]. The first is an isothermal analysis by using multiple measurements at different temperatures. However, isothermal analysis does not provide information on the reaction rate of the cross-linking but only on the degree of conversion at a given temperature, rendering it less useful to elucidate the reaction’s mechanism [10]. Furthermore, it tends to overlook temperature fluctuations during the curing, potentially leading to an underestimation or overestimation of the actual cure time, and it requires long observation times [10].

A second approach involves the use of DSC experiments employing different constant heating rates. Dynamic DSC tests provide insight into reaction kinetics by measuring the change in heat during the process. These experiments allow for the study of a cure system considering different temperatures and heating rates, thereby affording a deep understanding of the cross-linking mechanisms and the thermal characteristics of the material [11].

Two common approaches to study the curing kinetics using non-isothermal tests are “model-free” and “model-based” methods. The use and effectiveness of these methods have been well documented in the literature [10,12,13]. Model-free techniques, such as isoconversional analysis, provide insight into epoxy reactivity as a function of temperature without any assumptions about the reaction mechanisms [14]. These methods are based on the determination of the activation energy (*E_a_*) as a function of the degree of conversion (*α*) [15]. Currently, they are widely used to study the kinetics of complex reactions, including the cure of epoxy resins [16,17].

In contrast, model-based methods rely on the assumption of a specific reaction mechanism and employ mathematical models to describe the curing kinetics of epoxies [18]. Data-driven methods use experiments to collect data and statistical tools to analyse them.

The combined use of these two approaches yields detailed knowledge of the whole process, enabling better design and optimisation of thermoset materials for specific applications [19].

To conclude the study, tensile tests were performed to verify the thoroughness of the curing process and to assess the mechanical properties of the materials.

## 2. Materials and Methods

### 2.1. Epoxy–Novolac Systems

Novel bio-based novolac–epoxy matrices were formulated by blending commercially available products: two epoxies, one novolac resin and two amine hardeners have been purchased from Cardolite^®^, Specialty Chemicals Europe NV, Mariakerke (Gent), Belgium.

The epoxy resin, FormuLITE™ 2501A, is a bisphenol-type epoxy resin diluted with a CNSL-based reactive diluent. A bio-content of 34.3% and an epoxy equivalent weight (EEW) of 198 g/eq have been calculated.

2501A composition includes over 80% of 2,2-bis-[4-(2,3-epoxypropoxy)phenyl]-propane (BADGE) (CAS: 1675-54-3) and less than 20% of Cashew (Anacardium occidentale) Nutshell Extract, decarboxylated, distilled, oligomerisation products with 1-Chloro-2,3-epoxypropane (CAS: 68413-24-1).

The amine hardeners, FormuLITE™ 2401B and 2002B, are mixtures of aliphatic and cycloaliphatic amines with bio-based content of 33.0 and 67.7% and an EEW of 62 and 104 g/eq respectively.

The composition of 2401B includes less than 55% of 3-Aminomethyl-3,5,5-Trimethylcyclohexylamine (IPDA) (CAS: 2855-13-2), 12% of m-Phenylenebis(methylamine) (MXDA) (CAS: 1477-55-0), 4% of 2-Methylpentane-1,5-diamine (MPMD) (CAS: 15520-10-2), and 1% of 3,6,9-Triazaundecamethylenediamine (TETA) (CAS: 112-57-2). The remaining components are derived from CNSL technology. However, for privacy reasons, they are not disclosed by the supplier.

Similarly, 2002B is composed of less than 15% of m-Phenylenebis(methylamine) (MXDA) (CAS: 1477-55-0), 5% of 2-Methylpentane-1,5-diamine (MPMD) (CAS: 15520-10-2), and 1% of 3,6,9-Triazaundecamethylenediamine (CAS: 112-57-2). As with 2401B, the detailed composition of the bio-based components in 2002B is not provided by the supplier due to proprietary matters, leaving the remaining chemical composition unspecified.

According to Cardolite’s recommendations, 2501A epoxy resin shall be used in combination with the 2401B and 2002B hardeners following stoichiometric calculations based on the EEW values.

Technical datasheets, shown in Table 1 and Table 2, have been provided by the supplier.

Cardolite^®^ NC547 is a multifunctional epoxy–novolac resin containing a high bio-content (84.0% biomass) deriving from cardanol. It is a poly-glycidyl ether of an alkenyl phenol–formaldehyde–novolac resin with an EEW of 550–840 g/eq.

The NC547 chemical composition is composed of more than 99% Cashew Nut–Shell Liquid Epoxy (CAS: 115487-50-8) and less than 1% of Xylenes (CAS: 1330-20-7).

The cured resin has a lower elastic modulus, a lower glass transition temperature, and good flexibility.

These properties make NC547 advantageous in applications requiring enhanced flexibility, although the trade-off is typically a reduction in mechanical strength.

Moreover, Cardolite^®^ NC514 is a difunctional glycidyl ether epoxy resin based on CNSL technology.

The chemical composition of NC514 (65.0% bio-content) consists of 100% cashew, nutshell liquid, polymer with epichlorohydrin, and phenol (CAS: 68390-54-5). It has a chain of eight carbons separating the aromatic groups, which allows this resin to be used in conjunction with traditional epoxy resins to increase coating flexibility, abrasion resistance, and water and chemical resistance without affecting other properties.

Both NC547 and NC514 fall into the category of “modifiers for epoxy systems” and are employed to tailor material properties to specific application requirements, thereby improving performance and extending the life of the final products. In our specific case, the main intent of their use is the need to increase the percentage of bio-content within the final product and enhance material flexibility.

The use of these modifiers must be carefully balanced to avoid compromising the material’s mechanical properties, which is a key focus of this study.

At first, 20% of epoxy modifiers were blended with 2001A at 80%, followed by the addition of one of the two hardeners (2401B and 2002B), respectively, in stoichiometric amounts based on their EEW.

Samples without epoxy modifiers are also prepared to have a basic comparison and to understand how the addition of NC547 and NC514 affects the curing process.

The final mixtures obtained are reported in Table 3.

### 2.2. Differential Scanning Calorimetry

Differential scanning calorimetry (DSC) analysis with a TA Instruments (New Castle, DE, USA) mod. Q200 has been performed.

The Q200 can operate over a wide temperature range from ambient to 725 °C. With the addition of the RC690 refrigerated cooling system, the operational range of the Q200 is extended down to −180 °C, ensuring a stable thermal environment, which allows for the accuracy of kinetic data collected during the experiments.

Samples have been heated in an aluminium pan in a N_2_ atmosphere with a constant flow of 50 mL/min from −40 °C to 210 °C using heating rates of 2.5, 5.0, and 10.0 °C/min.

It has been documented in the literature that satisfactory results can be obtained by performing at least three measurements [11,23,24,25]

Considering the reaction mechanisms of thermosetting resins, the observed heat peak can be considered as the maximum value obtainable. To determine the baseline, it is generally recommended in the literature to use a linear interpolation approach between the two extreme points of the peak. Indeed, studies carried out on kinetic models under non-isothermal conditions indicate that using a straight line as a baseline between the two extreme points of the peak can give acceptable outcomes [26].

A second heating ramp has been considered to ensure complete cross-linking and determine the glass transition temperature (Tg) of the materials. Samples have been conditioned at −40 °C and then heated at a constant rate of 20 °C/min from −40 °C to 200 °C.

### 2.3. Kinetic Analysis

DSC data were processed with the Kinetics Neo software (NETZSCH-Gerätebau GmbH, Selb, Germany) to determine the exothermic heat generated during the curing reaction.

Kinetic analysis relies on mathematical models that describe the sequence of reactions over time, leveraging experimental data obtained through DSC.

Our investigation involves two different approaches: model-free and model-based techniques. These analytical methods result in kinetic models that accurately characterise experimental data over a range of temperature conditions.

The use of these methods allows for the prediction of the chemical system behaviour under a user-defined temperature range, offering the potential for process optimisation.

The model-based kinetic analysis possesses the capability to determine the number of reaction steps and the following parameters: enthalpy (exo/endothermic DSC effects), reaction type, activation energy, pre-exponential factor, and order of reaction.

#### 2.3.1. Model Free—Isoconversional Method

One of the fundamental hypotheses in model-free methods is represented by the dependence of the activation energy on the degree of conversion. The reaction function is assumed to be constant and unaffected by temperature and reaction rate [27].

Therefore, model-free allows for the determination of the activation energy without the assumption of reaction type using different points with the same conversion from DSC measurement at different heating rates [10]. These tools are built upon the concept that the reaction rates are temperature-dependent, enabling the study of reaction progression at different temperatures.

The integral form of the Arrhenius equation reveals how the reaction rate varies with temperature and yields essential kinetic parameters such as the activation energy (*E_a_*), the pre-exponential factor (*E*), and the reaction type [28].

Isoconversional methods are based on the following equation [29]:(1)$$\frac{d\alpha }{dt}={Ae}^{\frac{-Ea}{RT}}f(\alpha )$$
where *α* signifies the extent of conversion in the time (*t*), *A* is the pre-exponential factor, *E_a_* denotes the activation energy, *R* is the gas constant, *T* is the temperature, and *f*(*α*) is the model of the reaction. It operates by examining how the rate constantly changes with temperature for different degrees of conversion, providing insight into the complexity of the reaction and the temperature dependence of its mechanism.

Popular techniques, such as the rigid integral methods of Ozawa [30] and Flynn–Wall [31], employ strict integration limits, ranging from zero to a predetermined value of *α*, which leads to a systematic error in the estimation of *E_a_* when it shows large variations with *α* [10]. As explained by Sergey Vyazovkin et al. [12], to overcome this limitation, flexible integral methods have been introduced to allow integration over small *α*-segments. Among these, the differential Friedman method is highly recommended, especially for cross-linking processes, reducing the error and obtaining more accurate results.

#### 2.3.2. Model Fitting

Model-based methods are unique in their ability to anticipate polymerisation mechanisms by fitting dedicated reaction models to experimental data.

In our study, this approach was applied using the Kamal–Sourour (KS) reaction model.

Phenomenological kinetic analyses, based on general chemical models, are particularly effective [32]. In the context of amine–epoxy systems, these methods have been tailored to elucidate the chemical transformations intrinsic to these systems, due to the prevalence of epoxy-based thermosets.

The basis of all kinetic studies is the fundamental rate equation, which relates the rate of conversion at a constant temperature [33]:(2)$$\frac{d\alpha }{dt}=k\;f(\alpha )$$
where *α* is the chemical conversion, *k* is the rate constant, and *f(α)* is assumed to be temperature dependent.

Thermoset curing can be categorised into two types: n-th order and autocatalytic reactions. It is important to remember that curing is not necessarily limited to a single chemical reaction; often several reactions take place simultaneously or sequentially [31].

In the case of n-th order kinetics, the rate of conversion is proportional to the concentration of unreacted material (reactant concentration), expressed as follows:(3)$$\frac{d\alpha }{dt}={k\;(1-\alpha )}^{n}$$
where *n* is the reaction order.

Autocatalyzed thermoset cure reactions, where one of the reaction products acts as a catalyst for subsequent reactions, are mathematically represented as follows:(4)$$\frac{d\alpha }{dt}={k\;\alpha \;}^{m}{\;(1-\alpha )}^{n}$$
where *m* is another reaction order. Here, 1 − *α* is the concentration of reactants (epoxide and amine hydrogen), while *α* is the catalyst concentration (hydroxyl groups formed in the amine–epoxide reactions).

It is crucial to select the appropriate equation for interpreting thermoset cure data.

In many cases, particularly when the reaction mechanism is unknown, a combination of n-th order and autocatalytic equations is necessary, especially to account for the presence of hydroxyl groups [32].

In this field, the Kamal–Sourour model has been widely used in the literature [33,34,35] to understand complex polymerisation reactions.

This model [36], which is well established in the field of epoxy resin systems, combines the autocatalytic behaviour with the n-th order reaction model, resulting in the following equation [32]:(5)$$\frac{d\alpha }{dt}=\left(k_{1}+k_{2}\alpha^{m}\right){\left(1-\alpha \right)}^{n}$$
where *k*_1_ and *n* describe the n-th order reaction, while *k*_2_ and *m* are the autocatalytic contribution to the reaction [33,37]. Both *k*_1_ and *k*_2_ follow the Arrhenius equation.

Specifically, *k*_1_ represents the rate constant associated with a specific phase of the reaction. It indicates how quickly a specific part of the polymerisation process occurs under the influence of a catalyst (catalytic effect).

*k*_2_ *α^m^* is the autocatalytic effect. Here, *m* is a reaction order term and *α* is the conversion fraction. When *α* is low, this term is small, indicating a weak autocatalytic effect at the beginning of the reaction. When the reaction progresses, this term becomes more significant, indicating an increase in the autocatalytic effect in the advanced stages of the reaction [38].

(1 − *α*)*n* is the fraction of unreacted material; this term is high at the start of the reaction, indicating that most of the monomer is still present. As the reaction proceeds, this term decreases until the reaction is complete [32].

### 2.4. Specimen Manufacturing

Silicone (Easy Composites CS25 Condensation Cure Silicone Rubber) moulds, filled with epoxy–novolac–hardener mixtures, have been used for specimens’ preparation. The use of silicone for sample manufacturing is a widely accepted technique that allows for accurate control of specimen geometry and dimensions.

The specimens were prepared following the procedure described by Iadarola et al. [7] in accordance with ASTM D638-10 standards [22].

The curing cycle used was derived from the technical datasheets provided by Cardolite^®^, specifically tailored to the 2501A in combination with its corresponding hardener, by placing the specimens in an oven with a temperature program of four hours held at room temperature followed by two hours at 80 °C and two hours at 120 °C.

### 2.5. Tensile Test—Digital Image Correlation

Dog-bone specimens were tested using an MTS Landmark (Eden Prairie, MN, USA) servo-hydraulic testing machine (25 kN load cell) equipped with a Digital Image Correlation (DIC) system (LaVision GmbH, Göttingen, Germany) to evaluate the tensile properties of the manufactured resins under quasi-static (0.1 mm/s) conditions. A random speckle pattern was applied to the specimen surface to track the displacement of dog bones, allowing for better image detection and enabling the calculation of the material deformation [7] and induced displacement thanks to the DIC system.

The width and thickness of each specimen were measured with a digital calibre with a resolution of 0.01 mm.

Three specimens for each mixture were prepared for each test, ensuring robust and reliable data analysis.

LaVision software (LaVision GmbH, Göttingen, Germany, with LaVis 10) was used to reprocess the data.

## 3. Results and Discussion

### 3.1. Non-Isothermal Differential Scanning Calorimetry (DSC)

DSC curves (Figure 1a,c,e,g) reveal a single-step reaction for all formulations. This is evident by the presence of a single exothermic peak. Also, it is visible that the maximum peak of the reaction shifts towards higher temperatures as the heating rate increases. This shift has an impact on the rate of the chemical reaction, showing the dependence of the reaction rate on temperature, which is a significant factor in the Arrhenius equation.

Considering Equation (1), at higher temperatures the rate constant *k* increases. When the sample reaches the peak temperature, the chemical reaction occurs more rapidly (Figure 1b,d,f,h), resulting in a higher and typically hotter peak.

The presence of a single exothermic peak in the DSC curves suggests the possibility of a single-step reaction.

Nevertheless, visual inspection alone cannot always identify multi-step processes, and the absence of these features should not automatically indicate single-step kinetics. Indeed, deviations from the constancy of the activation energy, found as the degree of cross-linking increases, often suggest the involvement of multiple mechanisms in the polymerisation process [12]. Such deviations require further investigations to gain a holistic understanding of the process.

Furthermore, the interaction between epoxy and primary and secondary amines, as well as the etherification process involving epoxy and hydroxyl groups generated during the amine–epoxy reaction, is an interesting area of study: in most cases, primary and secondary amines exhibit comparable reactivity in aliphatic amines [32]. Consequently, the reactions with epoxide groups occur simultaneously, resulting in a single exothermic DSC peak during the curing process.

### 3.2. Isoconversional Analysis

A method to identify the correct kinetics involves determining the *E_a_* by examining the temperature dependence of its rate using the following isoconversional derivative of the overall rate [12]:(6)$$E_{a}=-R\;\;{\left[\frac{\;\partial ln\;(d\alpha /dt)\;}{\partial T^{-1}}\right]}_{\alpha }$$
where *α* is the degree of conversion at the time *t* while *T* is the reaction temperature.

*E_a_* represents a key parameter in the study of the curing process, and its constancy or insignificant variation is the main criterion in assessing whether the process can be treated as single or multi-step kinetics [12].

While in theory, *E_a_* should remain constant for a single-step reaction, in practise, it is never exactly constant. Therefore, the criterion of constancy is replaced by the criterion of insignificant variation, as clearly explained by Vyazovkin et al. [12]: understanding its fluctuations allows for optimising the resin formulation, controlling the conditions, and tailoring the final properties of the cured material.

Since parameter estimation in a kinetic model is based on *E_a_* values, its precision significantly affects the accuracy of the resulting parameters. Therefore, it is imperative to use the most accurate isoconversional methodology available, as discussed extensively in the ICTAC Kinetics Committee [10].

In this context, the use of flexible methods, such as the Friedman, is helpful in providing a reliable estimation of the cross-linking at different temperatures, which is important for the design of efficient production processes and the quality control of the final product. Also, Friedman analysis has been used to study the *E_a_* trends in the formulations.

Figure 2 compares the activation energy curves obtained from Friedman analysis for samples M1 and M1+20% NC547.

The activation energy for M1 increases immediately after the onset of the reaction and reaches a maximum of about 60 kJ/mol. This behaviour suggests the presence of an initial activation phase during which the reactants begin to interact with each other and form weak bonds.

The subsequent monotonically decreasing phase of the activation energy, between *α* = 0.10 and *α* = 0.85, shows how the curing reaction proceeds with lower order kinetics than the initial induction phase, making the formation of crosslinks between the molecules easier and faster.

The initial increase during the curing process can be attributed to several factors. One possible explanation is the presence of highly reactive species, which may have a higher activation energy due to specific chemical reactions involved, requiring a higher initial *E_a_* to overcome the reaction barriers and initiate the formation of the polymer crosslinks [39].

The subsequent almost linear decrease in activation energy from 61 kJ/mol to 41 kJ/mol indicates a change in the reactive mechanism or a transition to a diffusion-controlled regime. This decrease is attributed to a greater contribution of diffusion phenomena during the later stages of curing, where the availability of reactive species decreases and the diffusion of reactants through the polymer matrix becomes a limiting factor [40].

The increase in *E_a_* to approximately 46 kJ/mol when the resin cross-linking reaches 90% of conversion is related to additional processes occurring during this advanced curing stage. These processes could include the formation of secondary structures or collateral reactions, such as vitrification [41], which affect the *E_a_* required for the reactions.

The profile of the activation energy changes when novolac resin NC547 is added: in this case, *E_a_* starts at a value of ~52 kJ/mol, then it decreases relatively linearly until reaching 45 kJ/mol at α = 0.78 and then starts to exponentially increase up to 63 kJ/mol.

The initial decrease in *E_a_*, from 52 kJ/mol to 45 kJ/mol, is due to the introduction of NC547 novolac resin: it contains phenolic functionalities that can implement reactive cross-linking with epoxy resins, enhancing the reactivity of the whole system and leading to a more efficient conversion with a lower activation energy requirement for the initial stages of curing.

The subsequent exponential increase in *E_a_* beyond α = 0.80 can be associated with the occurrence of additional reactions and processes. As the curing proceeds, the network becomes more compact, and the system may exhibit a higher degree of steric hindrance. This limits the mobility of reactant molecules [42], and it can result in the formation of more complex structures and the involvement of secondary reactions, such as branching or side reactions, which require higher energy.

For M2 and M2+20% NC514, *E_a_* exhibits different behaviour, as shown in Figure 3.

For M2, *E_a_* starts relatively high, at 91 kJ/mol, at the beginning of the reaction and rapidly decreases to 60 kJ/mol, meaning that reactive species need a high energy to overcome the initial reaction barrier and to start the formation of the first crosslinks.

The subsequent phase of relatively stable *E_a_* values, ranging from 0.1 to 0.80 conversion, shows a transition to lower-order kinetics during the curing process. This phase suggests that the crosslinks between chains become progressively easier and faster.

The decrease from 61 to 56 kJ/mol is due to diffusion phenomena as curing advances [43], with reactant availability diminishing and the diffusion of reactants through the polymer matrix becoming a limiting factor.

The final increase in *E_a_*, up to 100 kJ/mol as the resin cross-linking approaches 100%, is once again due to the presence of additional processes such as the formation of secondary structures or collateral reactions (e.g., vitrification) [44], which may impact the *E_a_* required.

Considering the M2+20% NC514 profile (Figure 3), *E_a_* shows a distinct behaviour. It starts at a lower value, 40 kJ/mol, and gradually increases to 50 kJ/mol at a conversion of 0.5. This low initial value can be attributed to the introduction of NC514, which reacts with the base epoxy resin, enhancing reactivity and leading to a cross-linking process at lower *E_a_*.

However, at around a 0.85% conversion, *E_a_* undergoes a significant increase, reaching a value of about 71 kJ/mol. As in the previous case, when the cross-linking progresses, the system has limited mobility of reactants, leading to the formation of more complex structures that may reflect in higher *E_a_*.

The changes observed in the *E_a_* profile after the addition of NC514 reflect the interaction between the components of the system: the presence of the epoxy modifier introduces additional reactive sites, alters the kinetics of the cross-linking, and influences the formation of the polymer network [39].

The specific chemical composition of NC514, and particularly the nature and functionality of its phenolic groups, plays an important role in determining these observed variations in the *E_a_* profile.

To assess the constancy or insignificant variability of *E_a_*, it is recommended to analyse the values within the range of *α* = 0.1–0.9, where fluctuations are typically less pronounced [12].

The significance of variation in *E_a_* is evaluated by comparison with the uncertainty associated with its determination, typically ±5–10% of *E_a_*. To consider the variation as insignificant, the difference between the maximum and minimum values of *E_a_* should be less than ~20% from the average *E_a_* value.

As a result, in accordance with the recommendations provided by Vyazovkin et al. [12], we choose to approximate our systems as a single-step process: the variations in *E_a_* are within acceptable limits and do not significantly impact the overall reaction kinetics in the *α*-range considered.

The exact values of *E_a_* can be found in the Appendix A.

In summary, our findings align with the guidelines presented by Vyazovkin and support the classification (approximation) of the cure reaction as a single-step process, despite the presence of multiple components.

### 3.3. Kamal–Sourour

Kinetics Neo software from Netzsch Gerätebau GmbH facilitated model fitting by minimising the difference between measured and calculated values.

It is important to note that the applicability of Kamal–Sourour (KS) model is limited to the kinetic regime of the cross-linking process [12]: as the process goes forward, the cross-linking polymerisation often switches from the kinetic regime to the diffusion regime. This critical transition is characterised by a change in the activation energy, *E_a_*, leading to either abnormally low (<~40 kJ mol^−1^) or high (>~100 kJ mol^−1^) at higher values of *α* [12].

The uncured epoxy resins were heated at rates of 2.5, 5.0, and 10.0 °C/min. Based on experimental outcomes, exothermal curing was simulated with the KS 1-step reaction, and results are reported in Figure 4.

The model captured both autocatalytic and non-autocatalytic reactions, as demonstrated by the fitting of the model’s curves (Figure 4) and experimental conversion data.

The parameters extracted from the Kamal–Sourour model are reported in Table 4.

For M1, the KS method suggests the presence of two distinct reactions with different activation energies. The autocatalytic effect is noticeable, indicating that the reaction rate increases as the reaction progresses.

The addition of NC547 reduces the activation energy for both reactions, suggesting that it enhances the reactivity of the system, leading to a more efficient curing process.

In M2, the autocatalytic effect is even more pronounced, indicating a strong influence of the reaction products on the reaction rate.

Finally, the addition of NC514 enhances the autocatalytic effect, resulting in a lower *E_a_* for the first reaction. This indicates that this component promotes faster curing time. The high coefficient of determination (R^2^) shows the reliability of the model fits, which reinforces the accuracy of the obtained kinetic parameters.

The fitting of the Kamal–Sourour model Heat Flow vs. Temperature can be found in the Appendix A.

These insights are valuable for making predictions and optimising the curing processes for specific applications.

#### Model Curing Prediction

The use of Kinetics Neo software was used to provide predictive insights useful for real-world applications: it is possible to simulate and forecast the curing kinetics of our systems, M1, M1+20% NC547, M2, and M2+20% NC514, under various temperature profiles.

As shown in Figure 5, during the first 4 h at room temperature, M1 exhibited a linear conversion reaching *α* = 0.5. This rapid conversion is attributed to the low activation energy, allowing molecules to easily form bonds.

M1+20% NC547 exceeded the expectations, achieving ~0.7 of conversion. This increased reactivity can be explained by the catalytic effect of NC547. As shown in the *E_a_* plot (Figure 2), this sample has an initial *E_a_* of 52 kJ/mol, indicating a higher conversion rate than M1.

By increasing temperature to 80 °C, there is a visibly rapid increase in conversion during the first hour of the isothermal step for both M1 and M1+20% NC547. It is related to the higher thermal energy. At the molecular level, particles possess higher average kinetic energy due to the increased thermal motion. The higher thermal energy accelerates the chemical reaction involved in the cross-linking process. This acceleration is manifested in increased reaction rates.

The subsequent plateau observed during the second isothermal cycle indicates the onset of diffusion-controlled kinetics where reactant mobility becomes a limiting factor.

The final increase in temperature to 120 °C facilitates the completion of the reactions, resulting in 0.99 and 1.00 conversion for M1 and M1+20% NC547, respectively. At 80 °C, the diffusion of reactant molecules through the polymer matrix limits the progression of the curing reaction. However, by rising temperature to 120 °C, the thermal energy exceeds the *E_a_* barrier, allowing the reactant molecules to overcome the diffusion constraints. Consequently, the increased thermal energy accelerated the chemical reactions, allowing the cross-linking to be completed and the resin system to be fully cured.

For M2 and M2+20% NC514 (Figure 6), the slower initial conversion at room temperature, up to *α* = 30% and *α* = 40%, respectively, reflects the higher *E_a_* required for bond formation compared to the M1 series.

A rapid conversion occurs when the temperature is increased to 80 °C due to increased molecular collision and energy availability. However, the gradual decrease in conversion rate suggests a shift towards complex reaction pathways or steric hindrance as the reaction proceeds [45].

Again, increasing the temperature to 120 °C allows the reactions to reach completion, with conversionsup to *α* = 0.98 and *α* = 0.97 for M2 and M2+20% NC547, respectively.

These observed trends suggest a complex interplay between temperature, molecular mobility, and availability of reactive species.

The DSC curves of the cured materials, used to verify the curing process, can be found in the Appendix A.

### 3.4. Tensile Test

Stress–strain curves from quasi-static tensile tests of epoxy resins are shown in Figure 7. The curves show that the addition of NC547 and NC514 reduces the initial slope of the curve (representative of the elastic modulus), the Ultimate Tensile Strength (UTS), and the strain at failure of the base epoxy resin systems (M1 and M2, respectively).

This reduction is less evident for the M2 epoxy resin blend; the distinction is determined by several factors, including the lower total bio-content of the NC514 and the specific composition of the systems, resulting in better compatibility with the base M2.

Moreover, M1 blends are characterised by higher strain compared to M2 mixtures; indeed, they reach over 20% strain for the base epoxy system and over 15% after the addition of NC547 epoxy resin.

To assess the mechanical properties at increasing bio-content levels, numerical values concerning mechanical properties and the corresponding glass transition temperature (Tg) are reported in Table 5.

The UTS and Young modulus of the base M1 material are 52 MPa and 2291 MPa, respectively. The introduction of NC547 novolac resin in the epoxy blend leads to a significant reduction of about 32% and 35%, respectively. This phenomenon can be ascribed to the higher bio-content that led to a reduction in both Young Moduli and UTS. For the same reasons, M2 and M2+20% NC514 resins present higher values compared to M1 resins. In this case, the reduction in the mechanical properties for the Young modulus and UTS is, respectively, 20% and 12%, by increasing the total bio-content by 5%. Indeed, M2 epoxy blends show a less sensible reduction in the mechanical properties after the addition of NC514. Similarly, the increase in bio-content led not to a significant reduction in the glass transition temperature that is the same for both M1 and M2 resins, 3%.

## 4. Conclusions

This work demonstrated that bio-based epoxy materials are viable replacements for petroleum-based counterparts, enhancing sustainability and contributing to the reduction of CO_2_ emissions. Tensile tests and DSC analysis on cured materials showed that epoxy modifiers (NC547 and NC514), when used at low percentages (up to 20%), allow for maintaining competitive mechanical properties compared to conventional fossil-based epoxy systems while achieving a bio-content of up to 50%. This aligns with the stringent environmental regulations set by the European Commission for the automotive sector, where these materials are envisioned to be used.

Furthermore, this study highlights the advantages of conducting kinetic analyses to predict the curing behaviour of epoxy systems in scenarios where multiple components are involved and/or their composition remains undisclosed due to proprietary reasons. This approach offers an efficient alternative to theoretical analyses of chemical reactions in complex systems, ensuring proper curing under a defined temperature ramp. Advanced kinetic analysis techniques, such as the Friedmann and Kamal–Sourour models, have proven effective in accurately describing the cross-linking behaviour of all the epoxy formulations (M1, M1+20% NC547, M2, and M2+20% NC514). The Friedman model captured the energy associated with cross-linking, enabling the determination of necessary parameters for the Kamal–Sourour model, which was subsequently used to validate the curing processes under defined temperature conditions.

## Figures and Tables

**Figure 1 polymers-16-02499-f001:**
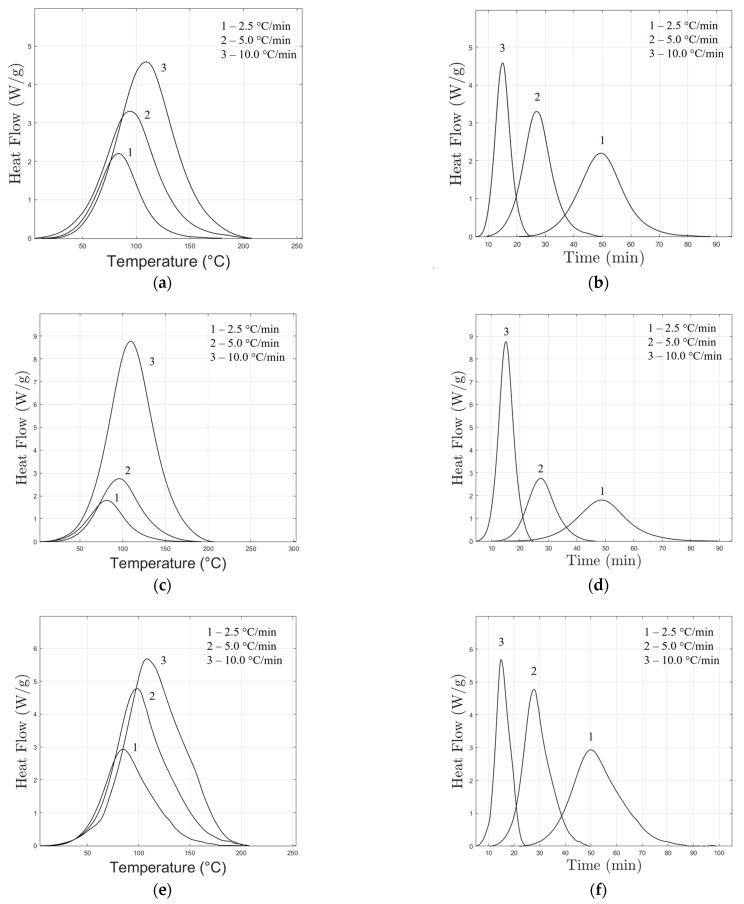
DSC curves vs temperature on the left and vs time on the right: (**a**,**b**) M1, (**c**,**d**) M1-20% NC547, (**e**,**f**) M2, and (**g**,**h**) M2-20% NC514 (exo-up).

**Figure 2 polymers-16-02499-f002:**
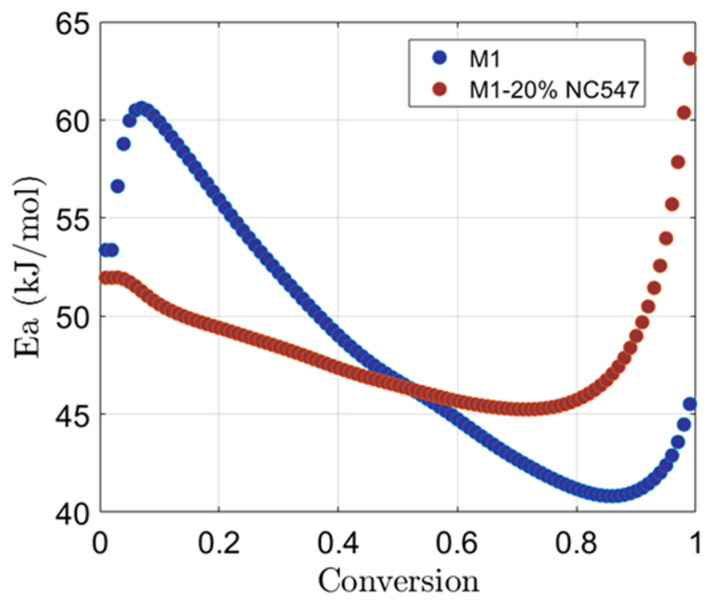
Activation energy vs. conversion for M1 and M1-20% NC547.

**Figure 3 polymers-16-02499-f003:**
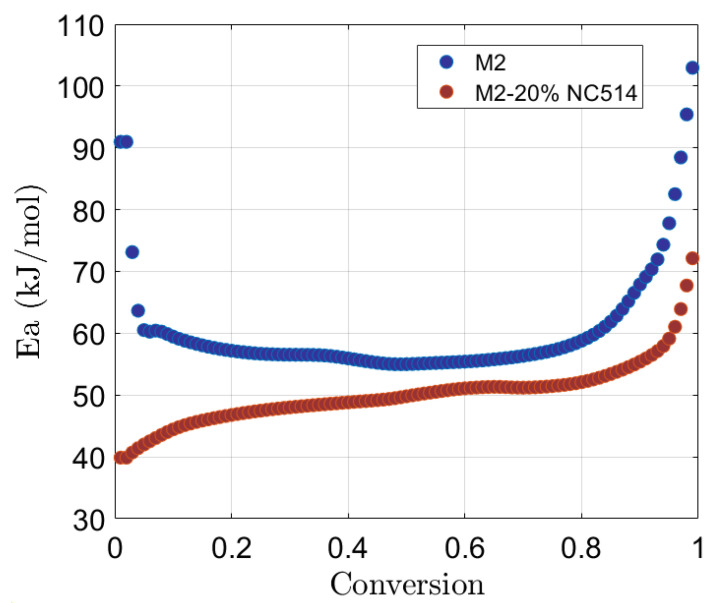
Activation energy vs. conversion for M2 and M2-20% NC514.

**Figure 4 polymers-16-02499-f004:**
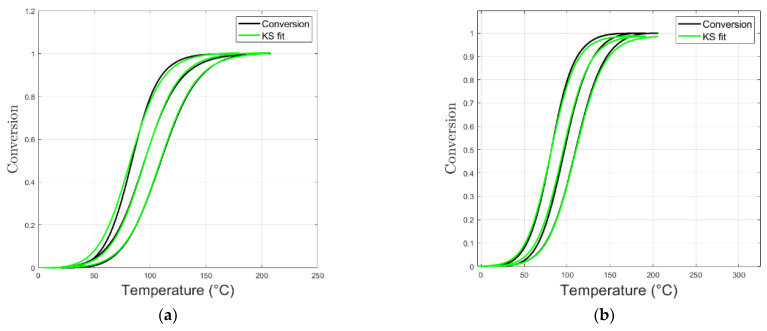
Conversion vs. temperature graphs (both experimental and simulated) for M1 (**a**), M1+20% NC547 (**b**), M2 (**c**), and M2+20% NC514 (**d**).

**Figure 5 polymers-16-02499-f005:**
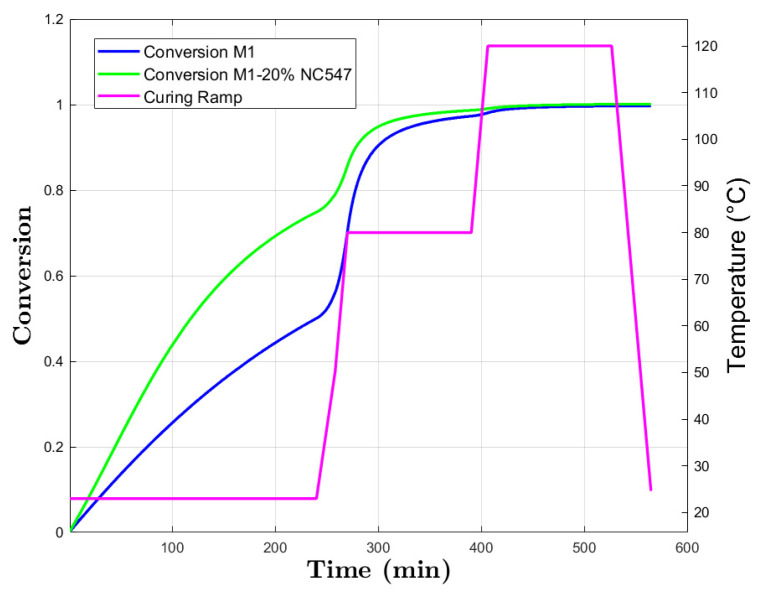
M1 and M1+20% NC547 conversion prediction and curing ramp.

**Figure 6 polymers-16-02499-f006:**
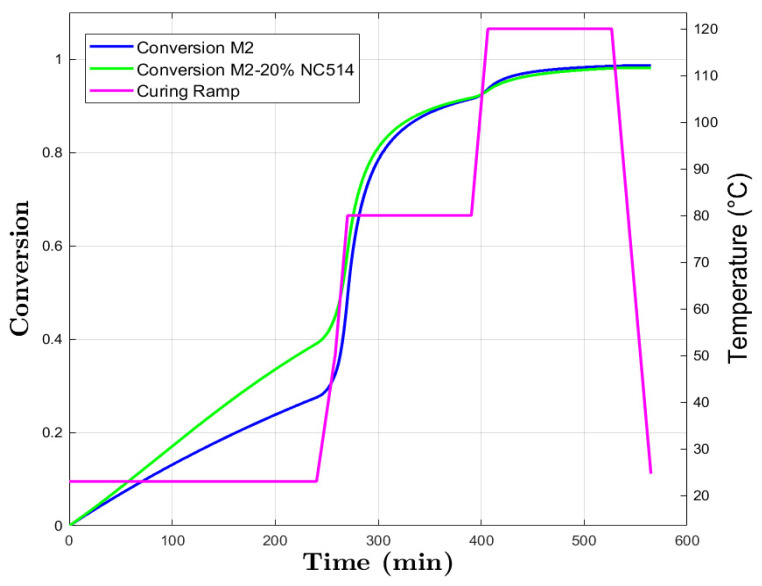
M2 and M2+20% NC514 conversion prediction and curing ramp.

**Figure 7 polymers-16-02499-f007:**
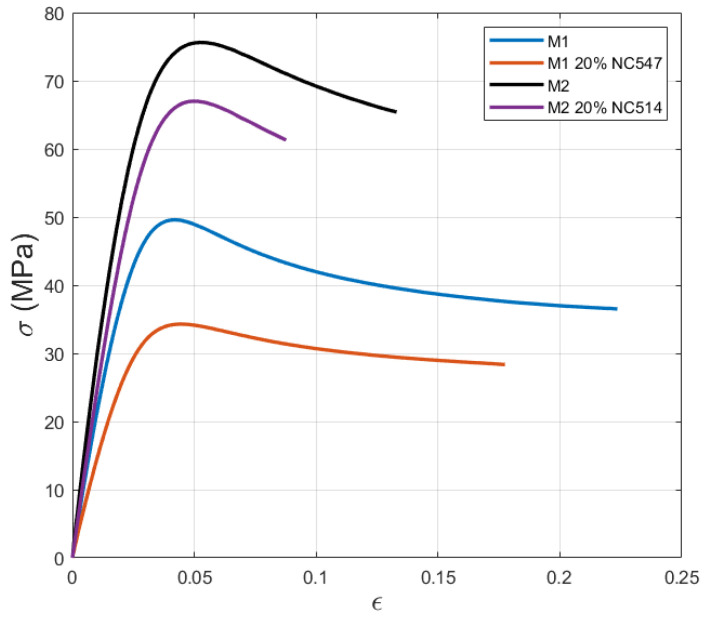
Stress–strain curves of tensile-tested materials.

**Table 1 polymers-16-02499-t001:** FormuLITE 2501A + 2002B properties.

Epoxy Resin 2501A + 2002B (M1)	Values	Test Method
Mixing ratio by mass	100:52	Internal supplier method
Mix Viscosity @ 25 °C [cPs]	1100	ASTM D2196 [20]
Pot life, 100 g mix @ 23 °C [min]	58	Internal supplier method
Glass transition temperature [°C]	73	ASTM 3418-99 [21]
Tensile Strength [MPa]	52	ASTM D638-10 [22]
Tensile Modulus [MPa]	2599	ASTM D638-10 [22]

**Table 2 polymers-16-02499-t002:** FormuLITE 2501A + 2401B properties.

Epoxy Resin 2501A + 2401B (M2)	Values	Test Method
Mixing ratio by mass	100:31	Internal supplier method
Mix Viscosity @ 25 °C [cPs]	905	ASTM D2196 [20]
Pot life, 100 g mix @ 23 °C [min]	95	Internal supplier method
Glass transition temperature [°C]	100	ASTM 3418-99 [21]
Tensile Strength [MPa]	69	ASTM D638-10 [22]
Tensile Modulus [MPa]	3134	ASTM D638-10 [22]

**Table 3 polymers-16-02499-t003:** Epoxy–novolac blends.

Name	Cardolite FORMULITE Blends	Bio-Content
M1	(2501A + 2002B) + 0% novolac	45%
M1-20% NC547	(2501A + 2002B) + 20% NC547	51%
M2	(2501A + 2401B) + 0% epoxy	34%
M2-20% NC514	(2501A + 2401B) + 20% NC514	39%

**Table 4 polymers-16-02499-t004:** Kamal–Sourour fitting parameters for epoxy–novolac blends.

	Fitting Parameters						
Sample	E_a_1 (kJ/mol)	Log Pre Exp (Log 1/s)	React Order (n)	Log Autocat PreExp (Log 1/s)	AutoCat Power (m)	E_a_2 (kJ/mol)	R^2^
M1	54.3	3.34	1.6	1.72	0.15	52.1	0.99807
M1-20% NC547	48.5	2.75	1.6	1.83	0.18	48.9	0.99960
M2	48.3	1.58	1.9	2.72	0.29	47.0	0.99504
M2-20% NC514	45.6	2.23	1.9	2.32	0.36	48.3	0.99884

**Table 5 polymers-16-02499-t005:** Bio-content, mechanical properties, and glass transition temperature values.

Sample	Total Bio-Content	UTS (MPa)	Elastic Modulus (MPa)	Tg (°C)
M1	45%	51	2291	65
M1+20% NC547	51%	34	1558	59
M2	34%	76	3098	92
M2+20% NC514	39%	67	2473	89

## Data Availability

The original contributions presented in the study are included in the article/Appendix A, further inquiries can be directed to the corresponding authors.

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
