# Peer review of "Cross-Linking Reaction of Bio-Based Epoxy Systems: An Investigation into Cure Kinetics"

_polymers, 2024, doi:10.3390/polym16172499_

Round 1

Reviewer 1 Report

Comments and Suggestions for Authors

I would like to extend my gratitude to the authors for their insightful research into study investigates the curing process of resin blends. The text of the paper is generally difficult to comprehend, which has prompted a number of queries regarding its relevance, in particular. Additionally, the paper is open to a few comments.

1. Lines 52-55 require further clarification. It would be beneficial to ascertain whether there are any other epoxy materials. It is unclear why the authors chose to highlight this particular material.

2. Line 56 states, "The production of novel epoxy systems using this approach requires". Please clarify the approach in question. Could you please clarify whether the authors are referring to cashew nut shell liquid (CNSL)? It is not the approach that is being claimed by the authors, but rather the material.

3.                                                                                                 It would appear that the authors are suggesting that a kinetic model is required in order to estimate the various curing parameters of a resin whose composition is currently unknown. The rationale behind this necessity was not elucidated by the authors. Lines 55–61 set out a number of parameters that must be determined. However, the rationale behind this necessity is not elucidated. It would appear that there is a semantic gap in the introduction, whereby the authors posit that rubber is an unfavourable material, and therefore it is necessary to create an environmentally-friendly alternative. One such alternative is cashew nut shell liquid (CNSL), which presents certain difficulties in terms of predicting the parameters associated with its use. Consequently, the authors argue that a model is required. The rationale behind the necessity of these parameters is not elucidated.

4. The rationale behind the description of the advantages and disadvantages of cardanol-based epoxy resin in the materials and methods section is unclear.

5. The abbreviation DSC should be deciphered when it first appears in a chapter. Here, the transcription is present after the first use of the abbreviation.

6. The description of the TA Instruments apparatus is required.

7.2.3.1 Model-free isoconversional method and 2.3.2 Model fitting. The authors provide an excess of detail on these items for the materials and methods section. The observations presented are of interest and contribute to a deeper understanding of the article. However, these observations are more appropriately situated in the results chapter, where the rationale and methodology behind the use of a particular formula can be elucidated. This would also facilitate a more coherent presentation of the results, as the reader can more readily comprehend the outcomes of this model without interruption.

8. The conclusion fails to provide an adequate account of the significance of the research conducted.

Comments on the Quality of English Language

The text presented is difficult to comprehend.

Author Response

Comment 1: [Lines 52-55 require further clarification. It would be beneficial to ascertain whether there are any other epoxy materials. It is unclear why the authors chose to highlight this particular material.]

Response 1: [Thank you for your observation and the opportunity to clarify. Our decision to use materials based on Cashew Nut Shell Liquid (CNSL) technology derived by our previous studies that showed promising mechanical properties in commercial products using this technology for automotive applications. In our research, we aimed to build on these findings by adjusting the mix and substituting some commercial products to achieve a better balance between mechanical properties and the bio-content of the final materials.

This choice ensures that the final epoxy materials are as sustainable as possible, meeting the European regulations for reducing emissions and promoting the use of renewable resources in automotive sector.

We recognize that this explanation should have been insert in the introduction, and we have now revised the manuscript to do so.

We hope this clarification addresses your concern and provides a clearer context for our work.]

Page 2 – Introduction – lines 57-60

[Among the various bio-based alternatives, Cashew Nut-Shell Liquid (CNSL) is regarded one of the most promising material due to its characteristics including excel-lent water and chemical resistance, a wide range of pot life and glass transition tem-perature, good reactivity at room temperature, high biological content, good processa-bility, and low cost [6].

Based on our previous studies, that demonstrates competitive tensile properties in cardanol-based compounds [7], this work focuses on different formulations in order to enhance the balance between bio-based content and mechanical performance to meet specific demands in the automotive industry.]

Comment 2: [Line 56 states, "The production of novel epoxy systems using this approach requires". Please clarify the approach in question. Could you please clarify whether the authors are referring to cashew nut shell liquid (CNSL)? It is not the approach that is being claimed by the authors, but rather the material.]

Response 2: [Thank you for the opportunity to clarify. The phrase "using this approach" refers to the method of blending various commercial resins together, not just the use of Cashew Nut Shell Liquid (CNSL) technology.

Indeed, the study focuses on the challenges of mixing epoxy and amine products, starting with the base epoxy resin 2501A and adding high bio-content resin like NC547 and NC514. This combination makes it difficult to predict how the curing process will behave, especially when considering whether the curing cycles recommended for the base resin 2501A, when used with hardeners 2002B and 2401B, would still be effective after adding these high bio-content resins.

So, by "this approach," we mean the process of combining multiple resins to increase bio-content while also maintaining specific material properties. In light of these matters, the kinetic study curing process was carried out to ensure that the ramp temperature used for the mixtures effectively cures the materials reaching the desired properties.

We have revised the introduction of the manuscript to make this explanation clearer.]

Page 2 – Introduction – lines 61-66

[Diverse commercial products have been blended: a bisphenol-type epoxy resin with a CNSL-based diluent, a multifunctional epoxy-novolac resin, a CNSL-based di-functional glycidyl ether epoxy resin, and two amine hardeners at varying bio content to develop novel epoxy products with bio-based content of up to 50%.

However, the production of new epoxy systems by blending various commercially available ones, requires a comprehensive study of the curing process: this is essential because the interaction between distinct resin components can give rise to unknown kinetics requiring a thorough analysis.]

Comment 3: [It would appear that the authors are suggesting that a kinetic model is required in order to estimate the various curing parameters of a resin whose composition is currently unknown. The rationale behind this necessity was not elucidated by the authors. Lines 55–61 set out a number of parameters that must be determined. However, the rationale behind this necessity is not elucidated. It would appear that there is a semantic gap in the introduction, whereby the authors posit that rubber is an unfavourable material, and therefore it is necessary to create an environmentally-friendly alternative. One such alternative is cashew nut shell liquid (CNSL), which presents certain difficulties in terms of predicting the parameters associated with its use. Consequently, the authors argue that a model is required. The rationale behind the necessity of these parameters is not elucidated.]

Response 3: [Firstly, we would like to clarify that our manuscript does not suggest that rubber is an unfavourable material. Rubber and epoxy resin have different applications, even though both are produced through a cross-linking process. Therefore, our focus was not on replacing rubber with epoxy resin, but rather on replacing conventional epoxy resins with bio-based alternatives, which is challenging due to the superior mechanical and thermal properties of the first.

The main goal we had was to highlight the significant challenges posed by conventional epoxy resins, primarily due to the toxicity of the components from which they are derived, particularly epichlorohydrin and bisphenol A, which have raised considerable environmental and health concerns.

Given that fully bio-based resins, while environmentally desirable, do not yet match the performance levels of conventional fossil-based resins, our research has been oriented toward developing intermediate solutions. These solutions involve combining fossil-based components with a certain level of bio-based diluent (2501A) and epoxy modifiers (NC547 and NC514), based on CNSL technology, to maximize the final bio-content while still maintaining good mechanical properties.

The necessity for a kinetic model arises from the complexities introduced by these novel and unknown formulations. Specifically, when blending different resins, hardeners and epoxy modifiers, particularly those with high bio-content, the curing process can be unpredictable due to the interactions between the various components. So, the intent behind employing a kinetic model is to understand how the curing process evolves in these complex systems.

When we refer to the need to estimate various curing parameters, we are specifically talking about determining the activation energy, as mentioned in lines 87 and 88 of the "Introduction" and then explained in the "Materials and Methods" section. The determination of activation energy is crucial because it provides insight into the energy associated with the curing process, which allows to approximate the number of "steps" in the curing reaction. We have endeavoured to explain this in detail in the "Model Free - Isoconversional Method" section, and it is discussed even more comprehensively in the article by Sergey Vyazovkin, "ICTAC Kinetics Committee Recommendations for Analysis of Multi-Step Kinetics," which we have cited in our bibliography.

In summary, the need to develop a kinetic model is important to determine if the curing ramp for these novel mixtures (for the best of our knowledge they are not documented in the literature) effectively ensures the correct cross-linking.]

Comment 4: [The rationale behind the description of the advantages and disadvantages of cardanol-based epoxy resin in the materials and methods section is unclear.]

Response 4: [By outlining both the benefits and potential limitations of cardanol-based epoxy resins, we aimed to justify their use for our experiments explaining how these materials align with the objectives of the European requirements.

Cardanol-based resins were mainly chosen for their high bio-content, favourable mechanical properties, and potential to reduce the environmental impact of epoxy systems. However, we also acknowledge the challenges associated with their use, particularly the difficulty in finding the right balance between achieving a high bio-content and maintaining the necessary mechanical properties in the final materials. This trade-off between bio-content and performance posed a significant challenge, as bio-based resins often do not exhibit the suitable properties required for specific applications.

In response to the feedback, we have adjusted the manuscript to ensure these points are more clearly articulated and to better explain the rationale behind our material choices.

We hope that these revisions will make the purpose and significance of our approach more evident.]

Page 2 – Introduction – lines 57-60

Page 4 – Materials and Methods – lines 151-158 and 167-168

Comment 5: [The abbreviation DSC should be deciphered when it first appears in a chapter. Here, the transcription is present after the first use of the abbreviation.]

Response 5: [Thank you for bringing this to our attention. We have corrected the oversight.]

Page 8 – Results and Discussion – line 314

Comment 6: [The description of the TA Instruments apparatus is required.]

Response 6: [A more precise and detailed description of the Q200 TA Instruments DSC apparatus has been added to the manuscript.]

Page 5 – Materials and Methods – lines 57-60 line 179-182

[The Q200 can operate over a wide temperature range from ambient to 725 °C. With the addition of the RC690 refrigerated cooling system, the operational range of the Q200 is extended down to -180 °C, ensuring a stable thermal environment, which allows for the accuracy of kinetic data collected during the experiments.]

Comment 7: [7.2.3.1 Model-free isoconversional method and 2.3.2 Model fitting. The authors provide an excess of detail on these items for the materials and methods section. The observations presented are of interest and contribute to a deeper understanding of the article. However, these observations are more appropriately situated in the results chapter, where the rationale and methodology behind the use of a particular formula can be elucidated. This would also facilitate a more coherent presentation of the results, as the reader can more readily comprehend the outcomes of this model without interruption.]

Response 7: [Thank you for your suggestion. We have restructured the manuscript accordingly. We have reworked the explanations originally placed in the Materials and Methods section and integrated them into the Results and Discussion chapter. We hope that these changes enhance the clarity and flow of the manuscript.]

Page 9 – Results and Discussions – lines 327-332

[Nevertheless, the visual inspection alone cannot always identify multi-step pro-cesses, and the absence of these features should not automatically indicate single-step kinetics. Indeed, deviations from the constancy of the activation energy, found as the degree of cross-linking increases, often suggest the involvement of multiple mecha-nisms in the polymerization process [13]. Such deviations require further investiga-tions to gain a holistic understanding of the process.]

Page 10 – Results and Discussions – lines 340-361

[A method to identify the correct kinetics involves determining the Ea by examining the temperature dependence of its rate using the following isoconversional derivative of the overall rate [13]:

E_a=- R  [( ∂ln (dα/dt) )/(∂T^(-1) )]_α

(6)

Where α is the degree of conversion at the time t while T is the reaction temperature.

Ea represents a key parameter in the study of the curing process and its constancy or insignificant variation is the main criterion in assessing whether the process can be treated as single or multi-step kinetics [25].

While in theory, Ea should remain constant for a single-step reaction, in practice, it is never exactly constant. Therefore, the criterion of constancy is replaced by the criterion of insignificant variation as clearly explained by Vyazovkin et al. [13]: understanding its fluctuations allows for optimizing the resin formulation, controlling the conditions and tailoring the final properties of the cured material.

Since parameter estimation in a kinetic model is based on Ea values, its precision significantly affects the accuracy of the resulting parameters. So, it is imperative to use the most accurate isoconversional methodology available, as discussed extensively in the ICTAC Kinetics Committee [10].

In this context, the use of flexible methods, such as Friedman, is helpful in providing a reliable estimation of the cross-linking at different temperatures, which is important for the design of efficient production processes and the quality control of the final product. Also, Friedman analysis has been used to study the Ea trends in the formulations.]

 Page 12 – Results and Discussions – lines 448-454

 [Kinetics Neo Software from Netzsch Gerätebau GmbH facilitated model fitting by minimizing the difference between measured and calculated values.

It is important to note that the applicability of KS is limited to the kinetic regime of the cross-linking process [13]: as the process goes forward, the cross-linking polymerization often switches from the kinetic regime to the diffusion regime. This critical transition is characterized by a change in the activation energy, Ea, leading to either abnormally low (< ~40 kJ mol-1) or high (> ~100 kJ mol-1) at higher values of α [13]. The model captured both autocatalytic and non-autocatalytic reactions, as demonstrated by fitting of model's curves (Figure 4) and experimental conversion da-ta.

The parameters extracted from the Kamal-Sourour model are reported in Table 4.]

Comment 8: [The conclusion fails to provide an adequate account of the significance of the research conducted.]

Response 8: [Thank you for the feedback. We have revised the conclusion section of the manuscript to better explain the significance of the results.]

Page 17 – Conclusions – lines 551-569

[This work demonstrated that bio-based epoxy materials are viable replacements for petroleum-based counterparts, enhancing sustainability and contributing to the reduction of CO2 emissions. Tensile tests and DSC analysis on cured materials showed that epoxy modifiers (NC547 and NC514), when used at low percentages (up to 20%), allow for competitive mechanical properties compared to conventional fossil-based epoxy systems, while achieving a bio-content of up to 50%. This aligns with the stringent environmental regulations set by the European Commission for the automotive sector where these materials are envisioned to be used.

Furthermore, the study highlights the advantages of conducting kinetic analyses to predict the curing behavior of epoxy systems in scenarios where multiple components are involved and/or their composition remains undisclosed due to proprietary reasons. This approach offers an efficient alternative to theoretical analyses of chemical reactions in complex systems, ensuring proper curing under a defined temperature ramp. Advanced kinetic analysis techniques, such as the Friedmann and Kamal-Sourour models, has proven effective in accurately describing the cross-linking behavior of all the epoxy formulations (M1, M1+20%NC547, M2, and M2+20%NC514). The Fried-mann model captured the energy associated with cross-linking, enabling the determi-nation of necessary parameters for the Kamal-Sourour model, which was subsequently used to validate the curing processes under defined temperature conditions.]

Reviewer 2 Report

Comments and Suggestions for Authors

The article entitled "Cross-linking reaction of bio-based epoxy systems: an investigation into cure kinetics" deals with the kinetics study of the curing process of bio based commercial resin (aromatic glycidyl ether based) and commercial amine hardener. 

DSC, and some model fitting methods were used to determine and confirm the kinetic of the curing process.

The article is interessant, and describe well the methods used for the kinetics of curing process.

Apart of this, the article is suitable for publication in Polymers.

Author Response

Comment 1: [The article entitled "Cross-linking reaction of bio-based epoxy systems: an investigation into cure kinetics" deals with the kinetics study of the curing process of bio based commercial resin (aromatic glycidyl ether based) and commercial amine hardener.

DSC, and some model fitting methods were used to determine and confirm the kinetic of the curing process.

The article is interessant, and describe well the methods used for the kinetics of curing process.

Apart of this, the article is suitable for publication in Polymers.]

Response 1: Thank you for your positive feedback and for recognizing the value of our work. We appreciate your comments and are glad that you found the methods and findings well-presented.

We are pleased that you consider the article suitable for publication in Polymers.

Reviewer 3 Report

Comments and Suggestions for Authors

1. The main remark is that this is a scientific journal and not an advertising one. When describing chemical reagents, it is necessary to talk about the chemical formula or provide the CAS number. In fact, without seeing the chemical structures it is difficult to predict any properties: aliphatic resins, cycloaliphatic or cyclic (if so, how many cycles). The description provided by the authors is insufficient. In the articles of the journal "Polymers" such descriptions are given (https://www.mdpi.com/2073-4360/16/15/2113)

2. The Conclusions state that a comprehensive study of epoxy systems for automotive applications has been conducted. Judging by the stress-strain curves, the authors’ polymers were not completely cured (tails after maximum strength) (Figure below). Authors are encouraged to provide DSC curves of the compositions. They will be used to judge the curing.

3. The references must be formatted according to the journal's requirements.

4. In Figures 1 b-h, it is necessary to enter the decoding for curves 1,2,3, as for Figure 1a

Author Response

Comment 1: [The main remark is that this is a scientific journal and not an advertising one. When describing chemical reagents, it is necessary to talk about the chemical formula or provide the CAS number. In fact, without seeing the chemical structures it is difficult to predict any properties: aliphatic resins, cycloaliphatic or cyclic (if so, how many cycles). The description provided by the authors is insufficient. In the articles of the journal "Polymers" such descriptions are given (https://www.mdpi.com/2073-4360/16/15/2113)]

Response 1: [Thank you for your comments. We understand the importance of providing comprehensive chemical information in scientific publications.

However, the materials we employed in our research are commercially available epoxy resins and hardeners, which are complex, proprietary formulations. Unlike pure chemical reagents, these products are composed of multiple components and designed to achieve specific performance characteristics. Therefore, they do not have a single CAS number or a single chemical structure that can be easily described. The proprietary nature of these formulations means that detailed chemical compositions are often not disclosed by manufacturers.

Moreover, in the article you referenced, the resins were synthesized by the authors from pure chemical reagents with known CAS numbers and chemical structures. This allowed the authors to provide the detailed chemical information you mean, which is appropriate and necessary for studies involving the synthesis of new compounds. In contrast, our study focuses on the application of commercially formulated products, which are already optimized for industrial use but could lack the transparency of detailed chemical composition due to their proprietary nature, and this was the reason why we decided not to include more information about it.

However, for some of the commercial product used (see 2501A, NC547 and NC514) we are in knowledge of all the component into the product and we are glad to report them  into the manuscriopt together with the relative CAS number. Regarding the two hardeners (see 2401B and 2002) only a percentage of the chemical component is disclosed by the supplier, however we have now reported what we know with their relative CAS number.

The complexity and proprietary nature of the formulations introduce a level of unpredictability in how the components will interact during the curing process when they are used to build novel mixtures, and this is the reason behind the choice to follow the path of a kinetic study instead of a theoretical approach of the reaction mechanism. Given the multiple components involved it would be hard to predict with certainty how the material will cure and cross-link based only on the general information provided by the manufacturers and this unpredictability is precisely why we regard a kinetic study necessary.]

Page 3 – Materials and Methods – lines 119-122

[2501A composition includes over 80% of 2,2-bis-[4-(2,3-epoxypropoxy)phenyl]-propane (BADGE) (CAS: 1675-54-3) and less than 20% of Cashew (Anacardium occidentale) Nutshell Extract, decarboxylated, dis-tilled, oligomerization products with 1-Chloro-2,3-epoxypropane (CAS: 68413-24-1).]

 Page 3 – Materials and Methods – lines 126-137

[The composition of 2401B includes less than 55% of 3-Aminomethyl-3,5,5-Trimethylcyclohexylamine (IPDA) (CAS: 2855-13-2), less than 12% of m-Phenylenebis(methylamine) (MXDA) (CAS: 1477-55-0), less than 4% of 2-Methylpentane-1,5-diamine (MPMD) (CAS: 15520-10-2), and less than 1% of 3,6,9-Triazaundecamethylenediamine (TETA) (CAS: 112-57-2). The remaining compo-nents are derived from CNSL technology. However, for privacy reasons, they are not disclosed by the supplier.

Similarly, 2002B is composed of less than 15% of m-Phenylenebis(methylamine) (MXDA) (CAS: 1477-55-0), less than 5% of 2-Methylpentane-1,5-diamine (MPMD) (CAS: 15520-10-2), and less than 1% of 3,6,9-Triazaundecamethylenediamine (CAS: 112-57-2). As with 2401B, the detailed composition of the bio-based components in 2002B is not provided by the supplier due to proprietary matters, leaving the remain-ing chemical composition unspecified.]

 Page 4 – Materials and Methods – lines 149-150, 153-154, 157-158, 167-168

[NC547 chemical composition is composed by more than 99% of Cashew Nutshell Liquid Epoxy (CAS: 115487-50-8) and less than 1% of Xylenes (CAS: 1330-20-7)…

…These properties make NC547 advantageous in applications requiring enhanced flexibility, although the trade-off is typically a reduction in mechanical strength…

…The chemical composition of NC514 consists of 100% Cashew, nutshell liquid, polymer with epichlorohydrin and phenol (CAS: 68390-54-5)…

…The use of these modifiers must be carefully balanced to avoid compromising the material’s mechanical properties, which is a key focus of this study.]

Comment 2: [The Conclusions state that a comprehensive study of epoxy systems for automotive applications has been conducted. Judging by the stress-strain curves, the authors’ polymers were not completely cured (tails after maximum strength) (Figure below). Authors are encouraged to provide DSC curves of the compositions. They will be used to judge the curing.]

Response 2: Thank you for bringing this to our attention. DSC curves have already been included in the Supplementary Information section of our manuscript. These curves provide the analysis of the curing process and confirm the extent of curing in our epoxy systems (Supplementary Material, Figure S1).

We hope this information addresses your concern, and we are confident that the DSC data will provide the necessary evidence to evaluate the curing of our materials.

Comment 3: [The references must be formatted according to the journal's requirements.]

Response 3: Thank you for identifying this oversight. We have corrected the references following the journal's formatting requirements.

Page 18 – References – lines 593-669

Comment 4: [In Figures 1 b-h, it is necessary to enter the decoding for curves 1,2,3, as for Figure 1a]

Response 4: We have now included legends for all the curves in Figures 1b-h.

Page 8 – Results and Discussion – lines 315

Round 2

Reviewer 3 Report

Comments and Suggestions for Authors

The authors have made the necessary clarifications to the comments. The corrected version of the article can be recommended for publication in the journal «Polymers»